# Energy imbalance alters Ca$^{2+}$ handling and excitability of POMC neurons

Lars Paeger[1,2†], Andreas Pippow[1,2†], Simon Hess[1,2], Moritz Paehler[1,2], Andreas C Klein[1,2], Andreas Husch[1,2], Christophe Pouzat[3], Jens C Brüning[2,4,5], Peter Kloppenburg[1,2*]

[1]Biocenter, Institute for Zoology, University of Cologne, Cologne, Germany; [2]Cologne Excellence Cluster on Cellular Stress Responses in Aging-Associated Diseases, University of Cologne, Cologne, Germany; [3]MAP5 - Mathématiques Appliquées à Paris 5, CNRS UMR 8145, Paris, France; [4]Department of Mouse Genetics and Metabolism, Institute for Genetics, Center of Molecular Medicine Cologne, Center for Endocrinology, Diabetes and Preventive Medicine, University Hospital of Cologne, Cologne, Germany; [5]Max Planck Institute for Metabolism Research, Cologne, Germany

*For correspondence: peter.kloppenburg@uni-koeln.de

†These authors contributed equally to this work

Competing interests: The authors declare that no competing interests exist.

**Abstract** Satiety-signaling, pro-opiomelanocortin (POMC)-expressing neurons in the arcuate nucleus of the hypothalamus play a pivotal role in the regulation of energy homeostasis. Recent studies reported altered mitochondrial dynamics and decreased mitochondria- endoplasmic reticulum contacts in POMC neurons during diet-induced obesity. Since mitochondria play a crucial role in Ca$^{2+}$ signaling, we investigated whether obesity alters Ca$^{2+}$ handling of these neurons in mice. In diet-induced obesity, cellular Ca$^{2+}$ handling properties including mitochondrial Ca$^{2+}$ uptake capacity are impaired, and an increased resting level of free intracellular Ca$^{2+}$ is accompanied by a marked decrease in neuronal excitability. Experimentally increasing or decreasing intracellular Ca$^{2+}$ concentrations reproduced electrophysiological properties observed in diet-induced obesity. Taken together, we provide the first direct evidence for a diet-dependent deterioration of Ca$^{2+}$ homeostasis in POMC neurons during obesity development resulting in impaired function of these critical energy homeostasis-regulating neurons.

## Introduction

Energy homeostasis is tightly regulated by highly dynamic neuronal networks in the hypothalamus. These control circuits adapt food intake and energy expenditure to the needs of the organism and the availability of fuel sources in the periphery of the body (*Apovian, 2016*; *Gao and Horvath, 2007*; *Power, 2012*). Dysregulation of this functional circuitry can cause metabolic disorders including obesity and type two diabetes, whose prevalence is increasing in Western societies (*Schwartz and Porte, 2005*). In the arcuate nucleus of the hypothalamus (ARH), satiety-signaling (anorexigenic), pro-opiomelanocortin (POMC)-expressing, and hunger-signaling (orexigenic) agouti-related peptide (AgRP)-expressing neurons integrate endocrine and metabolic factors to adapt neuronal activity that ultimately generates neurosecretory output (*Morton et al., 2006*; *Sohn et al., 2013*). This neurocircuitry not only regulates food intake and energy expenditure but also adapts glucose homeostasis to the acutely changing needs of the organism (*Belgardt et al., 2009*; *Blouet and Schwartz, 2010*; *Könner et al., 2007*; *Steculorum et al., 2016*).

Substantial progress has been made in identifying fuel-sensing endocrine and metabolic factors, such as leptin, insulin, glucose, free fatty acids, and uridine diphosphate (*Gao and Horvath, 2007*;

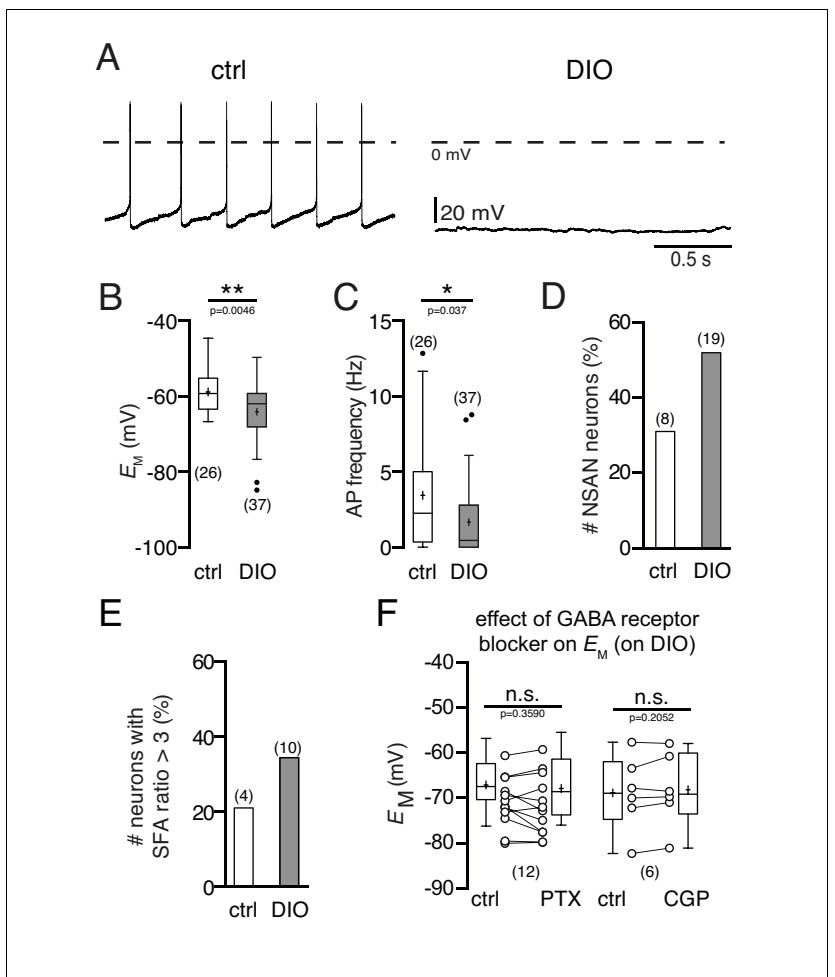

**Figure 1.** Diet-induced obesity decreases the spontaneous activity and hyperpolarizes the membrane potential of POMC neurons. Recordings were performed in the perforated patch-clamp configuration from eGFP expressing POMC neurons in the arcuate nucleus. (**A**) Original recordings of POMC neurons of mice on NCD and in DIO. (**B and C**) POMC neurons of DIO mice were hyperpolarized (**B**) and their action potential frequency was decreased (**C**) (Mann-Whitney test). (**D and E**) The percentage of silent (F < 0.5 Hz) POMC neurons (**D**) and the percentage of POMC neurons with strong SFA (SFA ratio >3 Hz) (**E**) was higher in DIO mice compared to controls. (**F**) The GABA$_A$ receptor blocker PTX and the GABA$_B$ receptor blocker CGP54626 did not restore the membrane potential of DIO mice to control (paired t-test). SFA, spike frequency adaptation. **p<0.01, ***p<0.001.

*Jordan et al., 2010*; *Steculorum et al., 2015*; *Varela and Horvath, 2012*), and many immediate actions of these hormones and nutrient components on POMC and AgRP neurons are well understood (*Claret et al., 2007*; *Jo et al., 2009*; *Parton et al., 2007*; *Spanswick et al., 1997*, *2000*).

In contrast, we have only limited information about the effects of sustained extreme nutritional states, such as high caloric intake and obesity, on the cellular electrophysiological properties and performance of these cells. It is well documented that increased caloric intake causes leptin and insulin resistance in ARH neurons (*Könner and Brüning, 2012*; *Varela and Horvath, 2012*), and that glucose sensing of POMC neurons is impaired in obesity (*Parton et al., 2007*). In POMC neurons, which are activated by reactive oxygen species (ROS), high-fat feeding promoted proliferation of peroxisomes and in turn reduced ROS levels and action potential firing (*Diano et al., 2011*). Increased expression of the suppressor of cytokine signaling (SOCS-3) has been shown to desensitize neurons to the anorexigenic actions of leptin and insulin (*Bjørbaek et al., 1998*; *Ueki et al., 2004*), and we and others have demonstrated that inflammatory signals activated by a high-fat diet cause neuronal leptin and insulin resistance (*Belgardt et al., 2010*; *Dietrich et al., 2013*;

*Kleinridders et al., 2009*; *Schneeberger et al., 2013*; *Tsaousidou et al., 2014*). Correspondingly, endoplasmic reticulum (ER) stress was recognized as an important causal factor for the development of leptin resistance (*Hosoi et al., 2008*; *Ramírez and Claret, 2015*). Most recent studies have identified mitofusin-2 (MFN2) as a direct link between ER stress and leptin resistance in the hypothalamus (*Schneeberger et al., 2013*). In anorexigenic POMC neurons diet-induced obesity (DIO) altered mitochondrial network dynamics and decreased mitochondria-ER contacts by downregulation of MFN2. It is important to note that this is a cell type-specific response, when compared to the functionally antagonistic orexigenic AgRP neurons in the ARH (*Dietrich et al., 2013*; *Schneeberger et al., 2013*). Since mitochondrial function is intimately linked to the dynamic behavior of these organelles, and since mitochondria play a crucial role in intracellular $Ca^{2+}$ signaling (*Szabadkai and Duchen, 2008*), we asked if intracellular $Ca^{2+}$-handling properties are modulated in POMC neurons under DIO.

This question is especially interesting, since several lines of evidence suggest that age-dependent changes in $Ca^{2+}$ homeostasis may partially increase, or even cause, the susceptibility for age-dependent impairment of neuronal function and neurodegeneration (*Berridge, 2012*; *Marambaud et al., 2009*; *Toescu and Verkhratsky, 2007*). Here, increased $Ca^{2+}$ load on the cell can result in neurotoxicity and impaired neuronal function (*Rizzuto et al., 2012*; *Rowland and Voeltz, 2012*; *Surmeier et al., 2010*). Thus, to assess whether $Ca^{2+}$ handling and the function of satiety-mediating POMC neurons are altered in diet-induced obesity, we analyzed the $Ca^{2+}$-handling properties of these neurons and their intrinsic electrophysiological characteristics in detail. We found that diet-induced obesity impaired intracellular $Ca^{2+}$ handling, including mitochondrial $Ca^{2+}$ uptake. The changes in $Ca^{2+}$ handling were accompanied by a marked decrease in activity and excitability of POMC neurons. Experimentally increasing or decreasing intracellular $Ca^{2+}$ concentrations reproduced the electrophysiological properties observed in diet-induced obesity.

## Results

### High-fat diet decreases activity of anorexigenic POMC neurons

To investigate the effects of diet-induced obesity (DIO) on the intrinsic electrophysiological characteristics of POMC neurons, we performed electrophysiological recordings on identified POMC neurons of 18-week-old transgenic mice expressing GFP under the control of the *Pomc* promoter (*Cowley et al., 2001*). The animals had been fed a normal chow diet (NCD) or a high-fat diet (HFD) for 12 weeks, starting at an age of six weeks. In an independent experiment we measured increase in body weight and body fat content during HFD exposure. Mice exposed to 11–13 weeks of HFD-feeding (starting at an age of 6 weeks) showed an elevation in body weight (control: 29.3 ± 1.9 g, n = 25; DIO: 38.6 ± 4.1 g, n = 25; p<0.0001, Mann-Whitney test) and body fat content (control: 12.7 ± 4.2%, n = 25; DIO: 32.1 ± 4.0%, n = 25; p<0.0001, Mann-Whitney test) compared to NCD-fed controls.

Using perforated patch recordings, the integrity of intracellular components was ensured. The activity of POMC neurons of mice exposed to a HFD was clearly reduced, which is in line with previous results (*Jo et al., 2009*). The membrane potential was hyperpolarized (*Figure 1A,B*; control: −58.8 ± 1.1 mV, n = 26; DIO: −64.0 ± 1.2 mV, n = 37; p=0.0046, Mann-Whitney test) and the spontaneous activity was decreased (*Figure 1A,C*; control: 3.4 ± 0.7 Hz, n = 26; DIO: 1.7 ± 0.4 Hz, n = 37; p=0.037, Mann-Whitney test). Moreover, the percentage of POMC neurons without spontaneous action potential firing (NSAN, not spontaneously active neuron; F < 0.5 Hz) increased from 31% (8 of 26) in control mice to 51% (19 of 37) in DIO mice (*Figure 1D*). Likewise, the number of POMC neurons with strong spike frequency adaptation was markedly enhanced, i.e., the percentage of neurons with a spike frequency adaptation ratio (SFA, for details see Materials and methods) >3 increased from 21% (4 of 19) in the control group to 34% (10 of 29) in the DIO group (*Figure 1E*).

Previous studies reported changes in both glutamatergic and GABAergic synaptic innervation of POMC neurons during a sustained HFD (*Klöckener et al., 2011*; *Newton et al., 2013*). Since all our experiments were performed in the presence of ionotropic synaptic blockers (for details see Materials and methods), it is unlikely that the observed hyperpolarization of the membrane potential in POMC neurons was caused by changes in synaptic input. However, to define the effect of potentially altered GABAergic input on POMC neurons in DIO, we performed two sets of experiments. Whole-

cell voltage-clamp recordings revealed that the inhibitory postsynaptic current frequency was increased in DIO. Application of picrotoxin (100 µM), a GABA$_A$ receptor antagonist, and CGP54626 (50 µM), a GABA$_B$ receptor antagonist, blocked GABAergic currents and eliminated IPSCs completely (data not shown). However, neither picrotoxin nor CGP54626 restored the membrane potential and firing rate to control levels (*Figure 1F*; PTX: n = 12, p=0.3590; CGP: n = 6; p=0.2052; paired t-tests). The results show that the DIO-induced hyperpolarization and the decrease in firing rate cannot be exclusively attributed to the increase in inhibitory input organization of these cells, but are also a consequence of altered cell-intrinsic properties.

Overall, we found that POMC neurons without spontaneous activity had a two-fold higher SFA ratio than spontaneously firing POMC neurons (*Figure 2A–C*; POMC firing: 2.0 ± 0.1, n = 27; POMC silent: 3.6 ± 0.3, n = 22, p<0.0001, Mann-Whitney test). This was particularly reflected in a strong reduction of the action potential number over the time course of the depolarizing stimulus

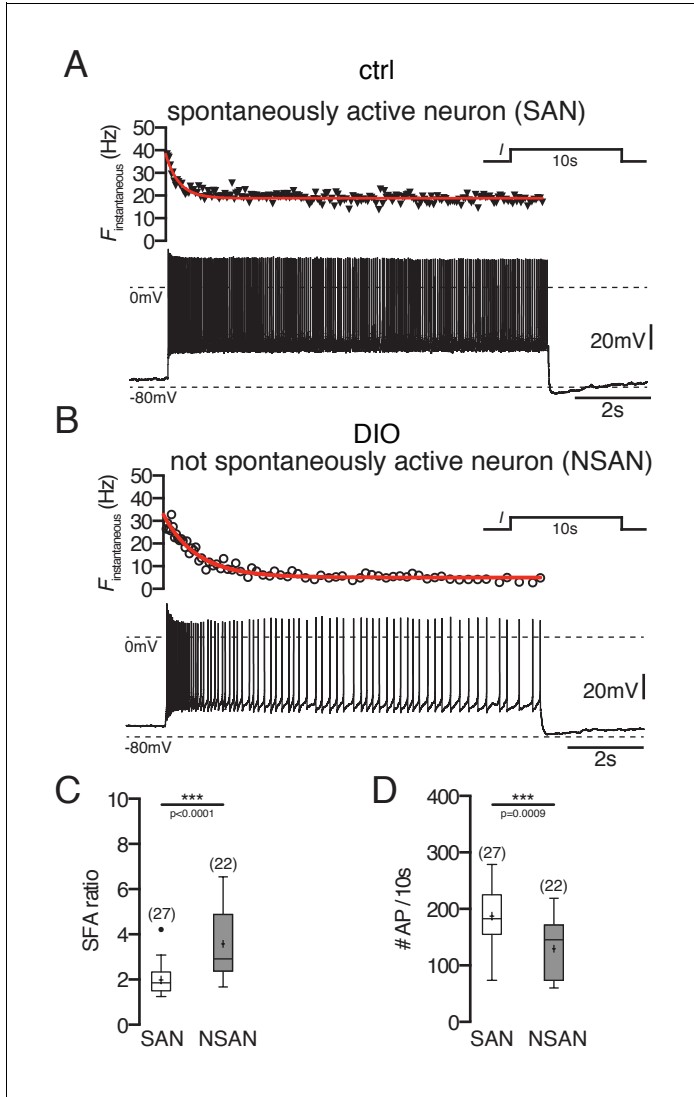

**Figure 2.** Spike frequency adaptation of POMC neurons is stronger in DIO mice. (**A and B**) Original recording (bottom) and the corresponding instantaneous spike frequency plot (top) of a spontaneously firing (**A**) and silent (**B**) POMC neuron during a 10 s depolarizing current injection. (**C and D**) SFA ratios are higher in silent POMC neurons (**C**) and, accordingly, the number of APs during the 10 s depolarization decreased (**D**) (Mann-Whitney test). SFA, spike frequency adaptation; SAN, spontaneously active neuron; NSAN, not spontaneously active neuron (F < 0.5 Hz). ***p<0.001.

(*Figure 2D*; POMC firing: 188 ± 10, n = 25; POMC silent: 130 ± 11.0, n = 22; p=0.0009, Mann-Whitney test).

Collectively, these data show that in DIO mice, the excitability of anorexigenic POMC neurons is markedly reduced.

## DIO alters Ca$^{2+}$ handling of POMC neurons

Further, we compared the Ca$^{2+}$-handling properties of POMC neurons, which represent a critical determinant of neuronal activity, of control and DIO mice, by using ratiometric Ca$^{2+}$ imaging with fura-2 (*Grynkiewicz et al., 1985*). Surprisingly, these experiments revealed increased levels of free intracellular Ca$^{2+}$ in POMC neurons in DIO compared to control mice (*Figure 3A*; control: 19.3 ± 2.3 nM, n = 11; DIO: 45.8 ± 7.3 nM, n = 13; p=0.0007, Mann-Whitney test), which was not expected, because of the DIO-induced hyperpolarization and decreased activity of the neurons.

Based on this finding, we next analyzed changes in Ca$^{2+}$-handling properties of POMC neurons. Since the dynamics of free intracellular Ca$^{2+}$ concentrations strongly depend on the endogenous Ca$^{2+}$-buffering capacity and the Ca$^{2+}$ extrusion rate, we also investigated whether DIO induces changes in the Ca$^{2+}$-handling parameters of POMC neurons by using the 'added buffer approach' in combination with whole-cell patch-clamp recordings and optical Ca$^{2+}$ imaging (*Neher and Augustine, 1992*; *Pippow et al., 2009*). The added buffer approach is based on a single compartment model with the rationale that for measurements of intracellular Ca$^{2+}$ concentrations with Ca$^{2+}$ chelator-based indicators, the amplitude and time course of the signals depend on the concentration of the Ca$^{2+}$ indicator (here: fura-2). The indicator acts as an exogenous Ca$^{2+}$ buffer and competes with the endogenous Ca$^{2+}$ buffer(s).

The kinetics of cytosolic Ca$^{2+}$ signals strongly rely on the endogenous and exogenous (added) Ca$^{2+}$ buffers of the cell. The amplitude and decay rate of free intracellular Ca$^{2+}$ changes with the increasing exogenous buffer concentration; the amplitude of free Ca$^{2+}$ decreases, and the time constant $\tau_{transient}$ of the decay increases, as shown in *Figure 3C and D*. If the buffer capacity of the added buffer is known, the time constant of the decay ($\tau_{transient}$; *Figure 3D*) can be used to estimate, by extrapolation, the Ca$^{2+}$ signal to conditions, with only endogenous buffers are present (-$\kappa_B$ = 1 + $\kappa_S$). The model used for this study (see *Equation 4*) assumes that the decay time constants $\tau_{transient}$ are a linear function of the Ca$^{2+}$-binding ratios ($\kappa_B$ and $\kappa_S$) (*Neher and Augustine, 1992*). The ratio $\kappa_S$ was determined from the negative x-axis intercept of the plot shown in *Figure 3E*. The slope of the fit is the inverse of the linear extrusion rate ($\gamma$). The point of intersection of the linear fit with the y-axis denotes the endogenous decay time constant $\tau_{endo}$ (no exogenous Ca$^{2+}$ buffer in the cell). To estimate the variability of these parameters, which were determined by linear fits, we used a bootstrap approach (n = 1000; as described in the Materials and methods). Using this method, we found that the endogenous Ca$^{2+}$-binding ratio (*Figure 3F*; control: 497 ± 18; DIO: 240 ± 5; p=0.0001, unpaired t-test), and, simultaneously, the Ca$^{2+}$ extrusion rate (*Figure 3G*; control: 150 ± 3 s$^{-1}$; DIO: 111 ± 1 s$^{-1}$; p<0.0001; unpaired t-test) were reduced in POMC neurons of DIO mice compared to control mice.

## DIO reduces mitochondrial Ca$^{2+}$ content of POMC neurons

Since mitochondria play a crucial role in intracellular Ca$^{2+}$ handling, we investigated whether the mitochondrial capacity of POMC neurons to accumulate Ca$^{2+}$ was affected in DIO mice (*Santo-Domingo and Demaurex, 2010*; *Szabadkai and Duchen, 2008*). This was achieved by determining the Ca$^{2+}$ release induced by the protonophore FCCP, as a measure for the ability of the mitochondria to contribute to the regulation of intracellular Ca$^{2+}$. Brain slices were AM-loaded with fura-2, and mitochondrial Ca$^{2+}$ release was induced in GFP-expressing POMC neurons by bath application of FCCP for 2 min (2 μM; *Figure 4A*). To monitor the Ca$^{2+}$ release, we recorded the fluorescence ratio (F340/F380) (*Figure 4B*), which is proportional to an increase in free cytosolic calcium ([Ca$^{2+}$]$_i$). FCCP induced a lower elevation of free cytosolic Ca$^{2+}$ in the DIO (Δ F340/F380 = 0.012 ± 0.001, n = 72 cells, N = 11 brain slices) than in the control cohort (Δ F340/F380 = 0.020 ± 0.003, n = 46 cells, N = 10 brain slices) (*Figure 4B,C*; p=0.0406, unpaired t-test). This indicates a reduced mitochondrial capacity to accumulate Ca$^{2+}$ in POMC neurons of DIO mice.

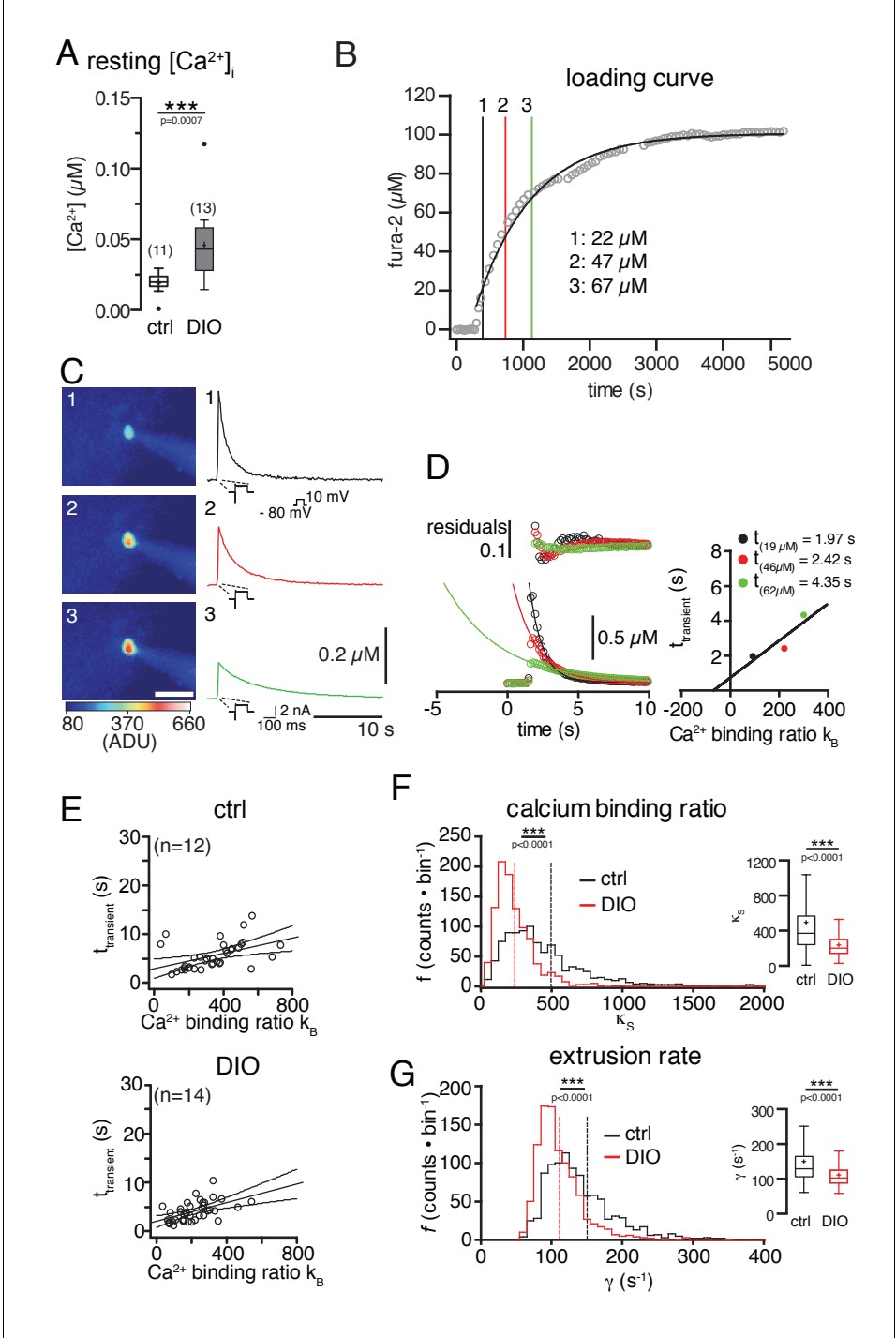

**Figure 3.** Diet-induced obesity changes the endogenous $Ca^{2+}$-handling properties of POMC neurons. $Ca^{2+}$ handling was analyzed using a combination of patch-clamp recordings, ratiometric $Ca^{2+}$ imaging, and the 'added buffer approach'. (**A**) $Ca^{2+}$ resting level. The concentration of free cytosolic $Ca^{2+}$ was increased in DIO mice. (**B–G**) $Ca^{2+}$-handling properties. (**B**) fura-2 loading curve. POMC neurons were loaded via the patch pipette with the ratiometric $Ca^{2+}$ indicator fura-2, which also serves as the added $Ca^{2+}$ buffer. Fura-2 fluorescence was acquired at 360 nm excitation (isosbestic point of fura-2) every 30 s, and converted into fura-2 concentrations. (**C**) Decay kinetics of voltage-induced $Ca^{2+}$ transients of the POMC neuron in (**B**). The images (left panels) were acquired at times indicated in (**B**) and demonstrate the increasing fura-2 concentration during loading. The graphs (right panels) demonstrate the effect of increasing added $Ca^{2+}$ buffer (fura-2) concentrations on the decay kinetics of voltage-evoked $Ca^{2+}$ transients. (**C and D**) Analysis of endogenous $Ca^{2+}$-handling parameters in a single cell. With

*Figure 3 continued on next page*

*Figure 3 continued*

increasing fura-2 concentrations, the amplitudes of transients decreased, and the time constants ($\tau_{transient}$) for decay were prolonged (**C and D**). The decay time constants were plotted against the $Ca^{2+}$-binding ratios of fura-2 ($\kappa_B$) (**D**). $\kappa_B$ was calculated from the intracellular fura-2 concentration, the $K_d$ of fura-2, and the resting concentration of free intracellular $Ca^{2+}$. The solid line represents the linear fit to the data. An estimate of $\kappa_S$ was obtained as the negative x-axis intercept. The $Ca^{2+}$ extrusion rate is estimated from the slope of the fit and the endogenous decay time constant from the intercept with the y-axis. (**E**) The decay time constants of all recorded neurons were plotted as a function of $\kappa_B$ for all POMC neurons of the control and DIO mice. The best linear fits with 95% confidence bands are shown. (**F and G**) To estimate the variance of the endogenous $Ca^{2+}$-binding ratio ($\kappa_S$) and the extrusion rate ($\gamma$), we used a bootstrap method (1000 samples), which provided bootstrap distributions (n = 1000) of the parameters for the control and DIO mice. Vertical lines indicate the means. (**F**) Distributions of $\kappa_S$. Sixteen counts for the control and one count for the DIO cohort between 2000 and 7500 are not shown. (**G**) Distribution of $\gamma$. Eighteen counts for the control and one count for the DIO cohort between 400 and 1350 are not shown. Subsequently, the distributions were log-transformed to bring them closer to a Gaussian, before applying unpaired t-tests. ***p<0.001.

## Spike frequency adaptation is increased in silent POMC neurons

Since the DIO-induced decrease in POMC neuron activity and excitability coincided with elevated levels of free intracellular $Ca^{2+}$, we hypothesized that both findings are linked and that the elevated $Ca^{2+}$ levels might cause the DIO-induced inhibition of POMC neurons. To verify this hypothesis, we

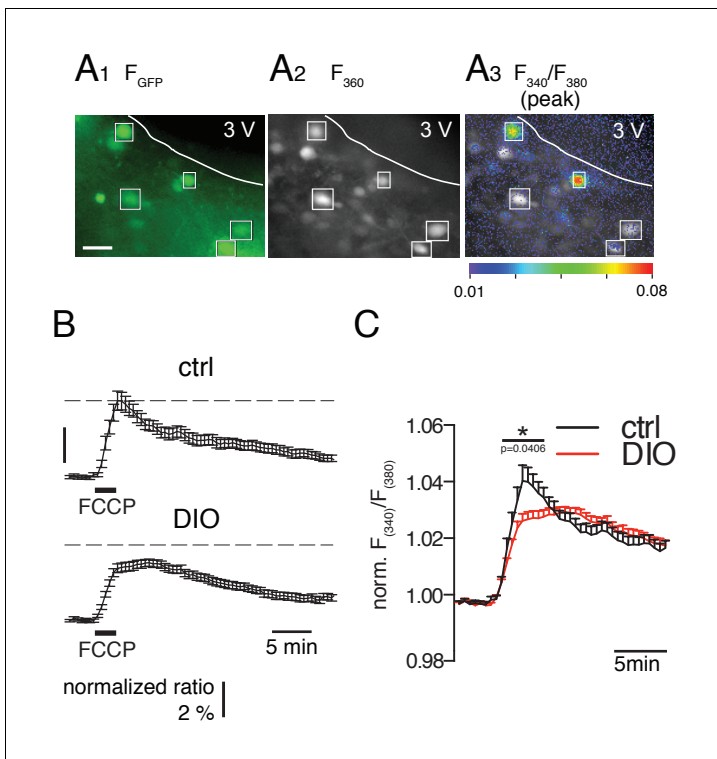

**Figure 4.** FCCP-induced mitochondrial $Ca^{2+}$ release is smaller in POMC neurons of DIO mice. (**A**) GFP-labeled POMC neurons (**A1**) and fura-2 fluorescence of AM-loaded neurons in a brain slice before (**A2**) and during (**A3**) FCCP (2 µM, 2 min) bath application. Regions of interests indicate double-labeled neurons, which were used for analysis. Scale bar: 50 µm. (**B**) Time course of FCCP-induced increase in intracellular $Ca^{2+}$ concentration [as $\Delta(F_{340}/F_{380})$ / $(F_{340}/F_{380})_0$] in POMC neurons of control and DIO mice. (**C**) For better visualization and comparison between the two populations, the fluorescence ratio of each recorded neuron was normalized to its baseline. The overlay in the right panel shows the release and uptake in higher resolution. Control, n = 46 cells, N = 10 brain slices. DIO, n = 72 cells, N = 11 brain slices. *p<0.05; Mann-Whitney test.

modified the intracellular Ca$^{2+}$ concentration by changing the extracellular Ca$^{2+}$ concentration and tested if the changes in intrinsic electrophysiological properties observed in DIO can be reproduced by changing the intracellular Ca$^{2+}$ concentration.

To confirm that changes in extracellular Ca$^{2+}$ concentration indeed affect free intracellular Ca$^{2+}$ levels, we combined electrophysiological recordings and ratiometric Ca$^{2+}$ imaging with fura-2. An example experiment is shown in *Figure 5A–C*. To prevent indirect (activity-induced) changes in intracellular Ca$^{2+}$ concentrations, the neurons were clamped at −70 mV with low-frequency voltage-clamp (implemented in an EPC10 patch-clamp amplifier, HEKA, Lambrecht, Germany). Elevating extracellular Ca$^{2+}$ concentrations from 2 mM to 4 mM increased intracellular Ca$^{2+}$ levels and decreased the hyperpolarizing holding current (*Figure 5B*). In contrast, reducing extracellular Ca$^{2+}$ caused a drop in free intracellular Ca$^{2+}$, increased the hyperpolarizing holding current, and even induced the generation of action potentials (*Figure 5A–C*). Using this direct approach to decrease or increase intracellular Ca$^{2+}$ levels, we investigated if the DIO-induced changes in excitability can be directly mimicked by adjusting the levels of free intracellular Ca$^{2+}$.

As hypothesized, reducing intracellular Ca$^{2+}$ depolarized the membrane potential and led to the generation of spontaneous action potentials in silent POMC neurons (*Figure 6A,C*). Increasing intracellular Ca$^{2+}$ reduced the action potential frequency in neurons that were spontaneously active (*Figure 6B,D*). It is important to note that this is not a general effect, that is, removing extracellular Ca$^{2+}$ did not change the intrinsic electrophysiological properties of hippocampal CA1 pyramidal cells of rats (*Penn et al., 2016*; *Su et al., 2001*). Reducing intracellular Ca$^{2+}$ concentrations in silent POMC neurons decreased the elevated SFA ratios to control levels of spontaneously active POMC neurons (*Figure 7A–C*). In neurons with spontaneous activity the SFA ratio decreased upon reducing the intracellular Ca$^{2+}$ concentration (*Figure 7C*). As a consequence of the reduced SFA, the number of action potentials per 10 s depolarizing current pulse was larger with low [Ca$^{2+}$]$_i$ (*Figure 7D*).

Moreover, we determined the current at which action potentials were generated (threshold current, $I_{threshold}$) by using ascending current ramps. In silent POMC neurons $I_{threshold}$ was higher compared to spontaneously active neurons, reflecting the decreased excitability of these cells

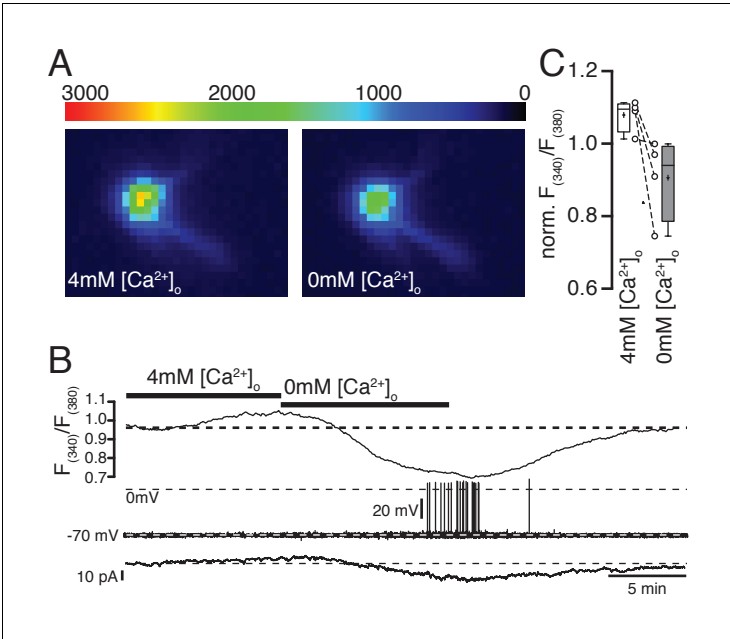

**Figure 5.** Ratiometric Ca$^{2+}$ imaging with fura-2 shows that increasing or decreasing extracellular Ca$^{2+}$ concentrations increases or decreases intracellular Ca$^{2+}$ levels, respectively. (**A,B,C**) Removal of extracellular Ca$^{2+}$ decreased intracellular Ca$^{2+}$ levels concomitant with an increase in hyperpolarizing holding current (**B**) when cells were clamped to −70 mV (low-frequency voltage-clamp). Note the generation of spontaneous action potentials in low Ca$^{2+}$ conditions. (**C**) Changes of intracellular Ca$^{2+}$ levels (normalized fura-2 ratios) under normal, high, and low Ca$^{2+}$ conditions (n = 4).

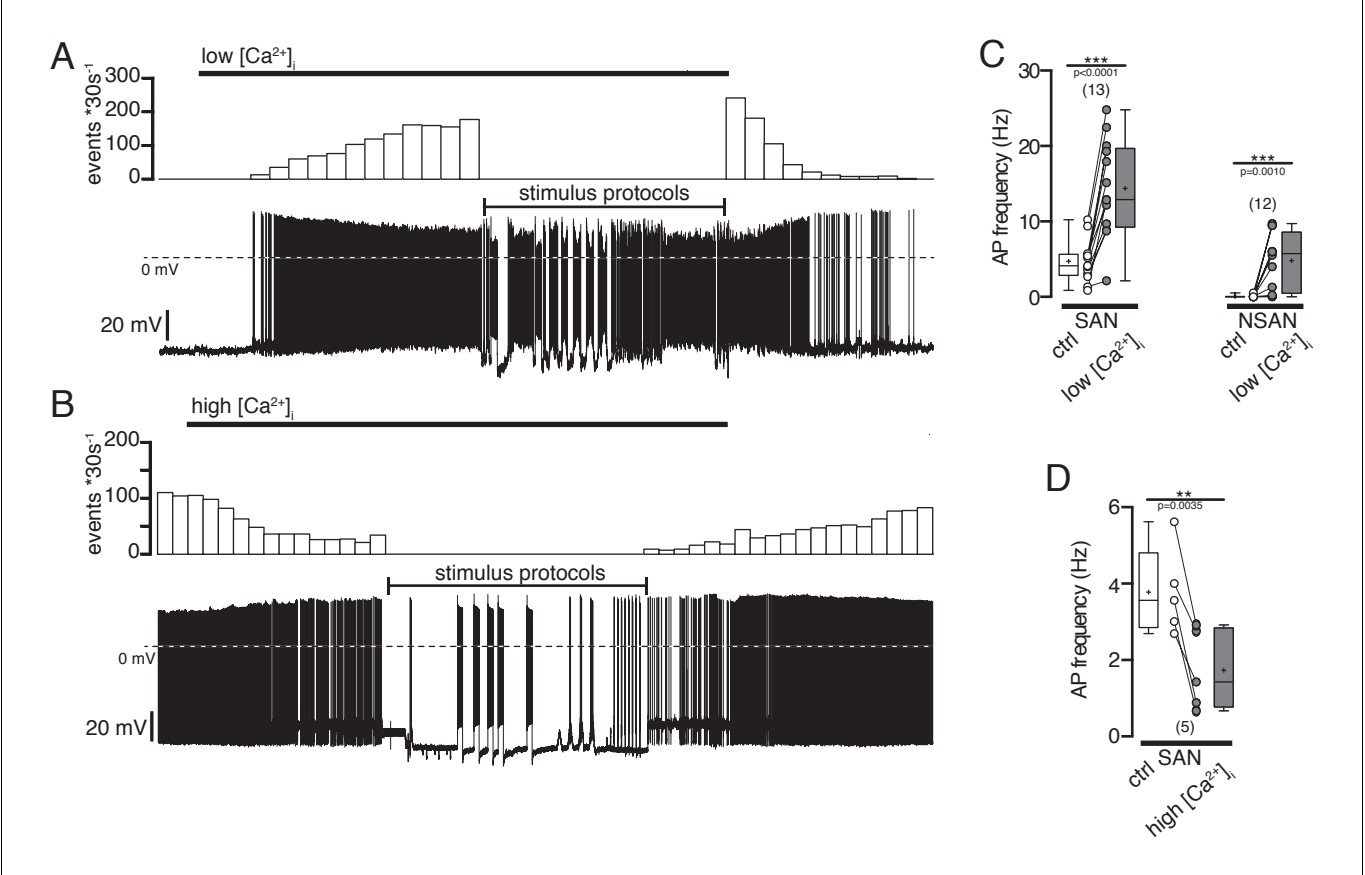

**Figure 6.** Effect of intracellular Ca²⁺ concentrations on action potential frequency of POMC neurons. (**A and B**) Original recordings of POMC neurons illustrating the effect of decreasing or increasing intracellular Ca²⁺ concentrations. (**A**) A silent POMC neurons started to generate action potentials with decreasing $[Ca^{2+}]_i$. The effect reverted to control levels at normal $[Ca^{2+}]_i$. (**B**) In a spontaneously active neuron, the AP frequency was reduced with increasing $[Ca^{2+}]_i$. The effect reversed at normal Ca²⁺ levels. (**C**) Action potential frequency of spontaneously active and silent POMC neurons at normal and low $[Ca^{2+}]_i$, respectively (paired t-test). (**D**) Action potential frequency of spontaneously active neurons at normal and high $[Ca^{2+}]_i$, respectively (paired t-test). SAN, spontaneously active neuron; NSAN, not spontaneously active neuron (F < 0.5 Hz). **p<0.01, ***p<0.001.

(*Figure 8A,B*). Decreasing $[Ca^{2+}]_i$ reduced $I_{threshold}$ in both firing and silent POMC neurons to similar values (*Figure 8A,B*). In line with these experiments, *current-frequency* relations were steeper in firing neurons compared to silent POMC neurons, and decreasing $[Ca^{2+}]_i$ increased the *current-frequency* relation to the same level in all cells (*Figure 8C*).

## SK channels contribute to decreased excitability of POMC neurons in DIO

Based on the findings that DIO increased the Ca²⁺ resting level and hyperpolarized the membrane potential, we hypothesized that activation of Ca²⁺-dependent K⁺ currents ($I_{K(Ca)}$) might contribute to the observed hyperpolarization and decreased neuronal activity. To verify this hypothesis, we used blockers for specific components of $I_{K(Ca)}$ and assessed if the changes in intrinsic electrophysiological properties, which we observed in silenced neurons and which were accompanied by elevated $[Ca^{2+}]_i$, can be reversed. We applied the toxins apamin, charybdotoxin, iberiotoxin, and paxilline, which have been shown to block specific components of $I_{K(Ca)}$ in vertebrates (*Bennett et al., 2000*; *Blatz and Magleby, 1986*; *Faber and Sah, 2003*; *Fioretti et al., 2004*; *Galvez et al., 1990*; *Ghatta et al., 2006*; *Giangiacomo et al., 1992*; *Kaczorowski et al., 1996*; *Pineda et al., 1992*; *Wolfart et al., 2001*; *Sanchez and McManus, 1996*; *Li and Cheung, 1999*; *Zhou and Lingle, 2014*). Apamin blocks SK (K$_{Ca}$ 2.1, K$_{Ca}$ 2.2, K$_{Ca}$ 2.3) channels, and iberiotoxin and paxilline blocks BK

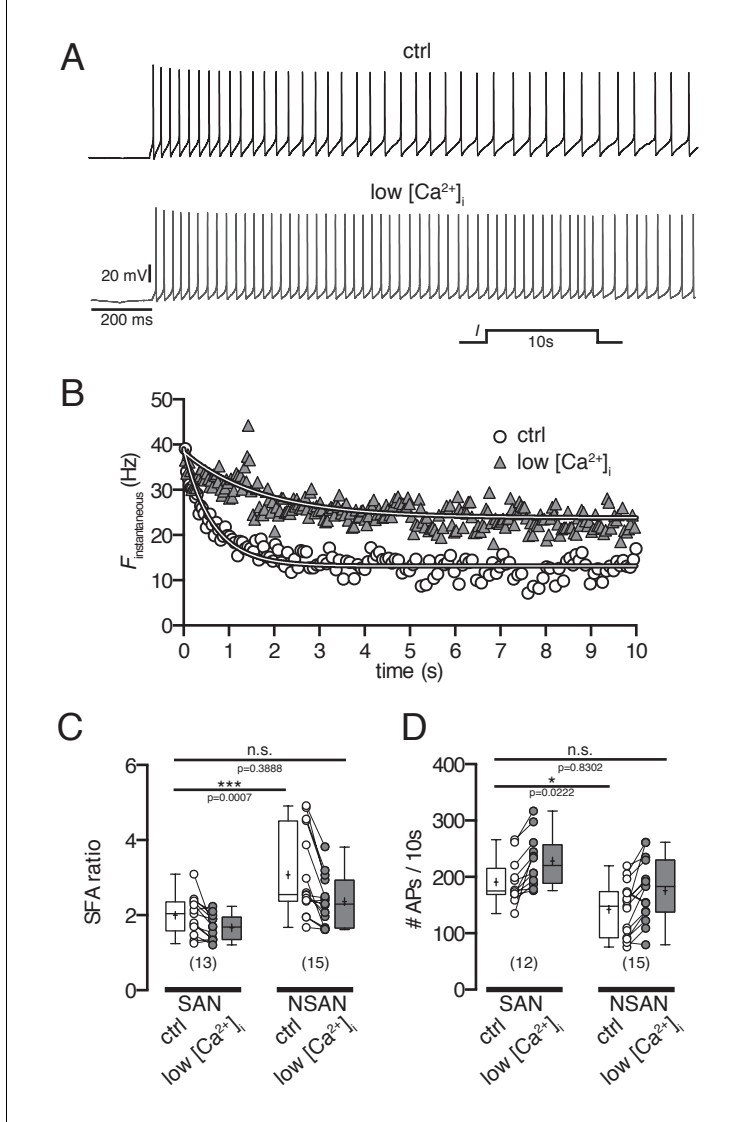

**Figure 7.** Spike frequency adaptation in POMC neurons depends on intracellular $Ca^{2+}$ concentrations. (**A and B**) Original recording (**A**) and corresponding plot of the instantaneous spike frequency (**B**) of a silent POMC neuron showing the SFA during a 10 s depolarizing current pulse at normal and reduced $[Ca^{2+}]_I$. Reducing $[Ca^{2+}]_i$ decreased the SFA. (**C**) SFA ratios of spontaneously firing and silent POMC neurons at different $[Ca^{2+}]_i$. In silent POMC neurons, the SFA is higher compared to firing POMC neurons. Reducing $[Ca^{2+}]_i$ in silent neurons decreases SFA ratios to levels of firing neurons (ANOVA with Bonferroni corrections). (**D**) The number of action potentials during the 10 s depolarizing current pulse is smaller in silent compared to firing POMC neurons. Reducing $[Ca^{2+}]_i$ increased the number of APs in silent neurons to the level of firing neurons (ANOVA with Bonferroni corrections). SFA, spike frequency adaptation; SAN, spontaneously active neuron; NSAN, not spontaneously active neuron (F < 0.5 Hz). *p<0.05, ***p<0.01.

($K_{Ca}$ 1.1) channels. Charybdotoxin not only blocks BK ($K_{Ca}$ 1.1) channels but has also been shown to block IK ($K_{Ca}$ 3.1) and $K_V$ ($K_V$ 1.2; $K_V$ 1.3) channels.

Bath-application of the BK channel blockers charybdotoxin (100 nM, n = 17), iberiotoxin (100 nM, n = 7) and paxilline (10 µM, n = 6) did not have reproducible effects on membrane potential, SFA or threshold currents to generate action potentials in silent, hyperpolarized POMC neurons (data not shown). In contrast, the SK channel blocker apamin (100–200 nM) had clear excitatory effects on these neurons, inducing depolarization and action potential firing in most of the silent POMC

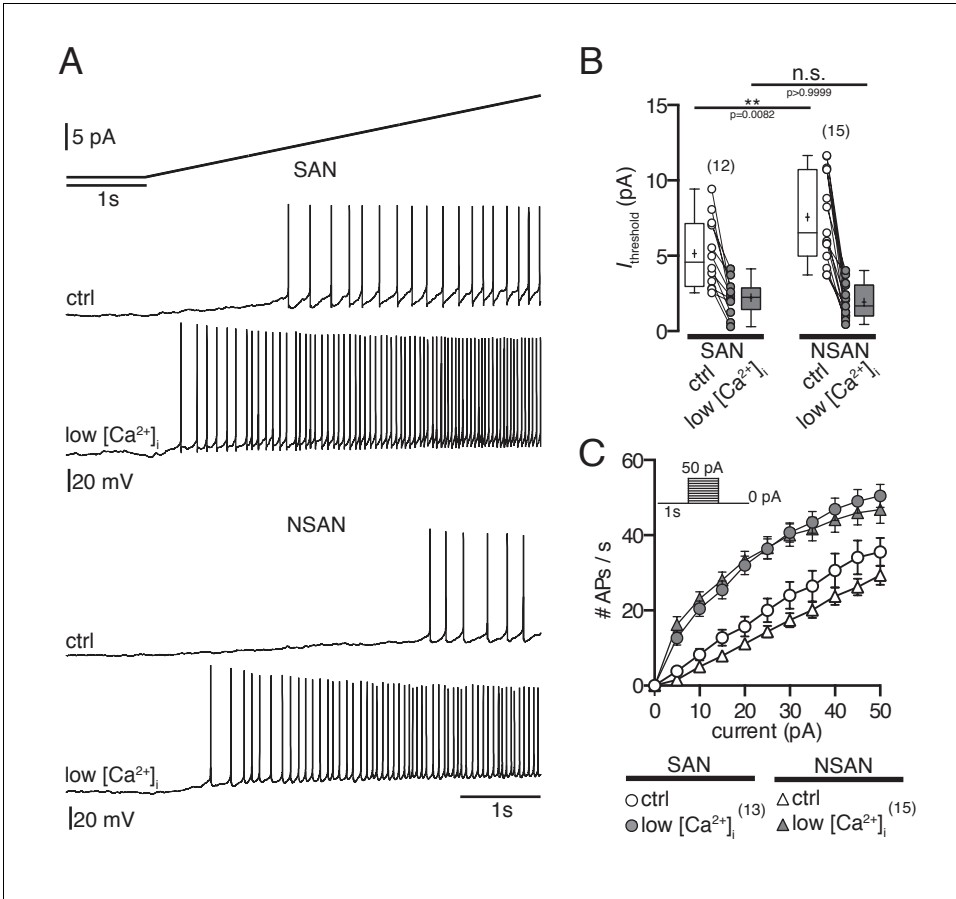

**Figure 8.** Low intracellular $Ca^{2+}$ concentrations increase excitability in POMC neurons. (**A and B**) Spontaneously active neurons have a lower threshold current than silent neurons to elicit action potentials. Reducing $[Ca^{2+}]_i$ decreased $I_{threshold}$ of silent and spontaneously active neurons to similar levels (ANOVA with Bonferroni analysis). (**C**) Number of action potentials during the current pulse as a function of injected current. Under control $[Ca^{2+}]_i$, the excitability is higher in spontaneously active neurons compared to silent POMC neurons (open symbols). Reducing $[Ca^{2+}]_i$ increased the excitability of all neurons to similar levels. SAN, spontaneously active neuron; NSAN, not spontaneously active neuron (F < 0.5 Hz). *p<0.05.

neurons (*Figure 9A–C*). Likewise, it reduced the SFA (*Figure 9D–F*) and decreased the threshold current for generation of action potentials in silent neurons, but not in spontaneously active control neurons.

Taken together, these blocker experiments suggest that SK channels are an import link between increased intracellular $Ca^{2+}$ levels and decreased excitability of POMC neuron in DIO. It is not surprising that the BK channel blockers did not depolarize silent neurons, since BK channels are highly voltage dependence (*Sah and Davies, 2000*) and might not contribute to the resting membrane potential. Interestingly, the SK channel blocker apamin, but not the BK channel blockers decreased SFA in silent neurons, suggesting that SK channels contribute markedly to SFA in silent POMC neurons, as shown for various cell types (*Brenner et al., 2005*; *Vandael et al., 2012*; *Yen et al., 1999*).

In summary, we show that DIO impairs the cellular $Ca^{2+}$-handling properties of POMC neurons and increases the resting level of free intracellular $Ca^{2+}$. In agreement with these general functional changes, we found the mitochondrial $Ca^{2+}$ content to be diminished. The changes in intracellular $Ca^{2+}$ handling properties were accompanied by membrane hyperpolarization and a marked decrease in excitability. By experimentally changing intracellular $Ca^{2+}$ concentrations we could reproduce the electrophysiological properties observed in DIO. Taken together, we provide the first

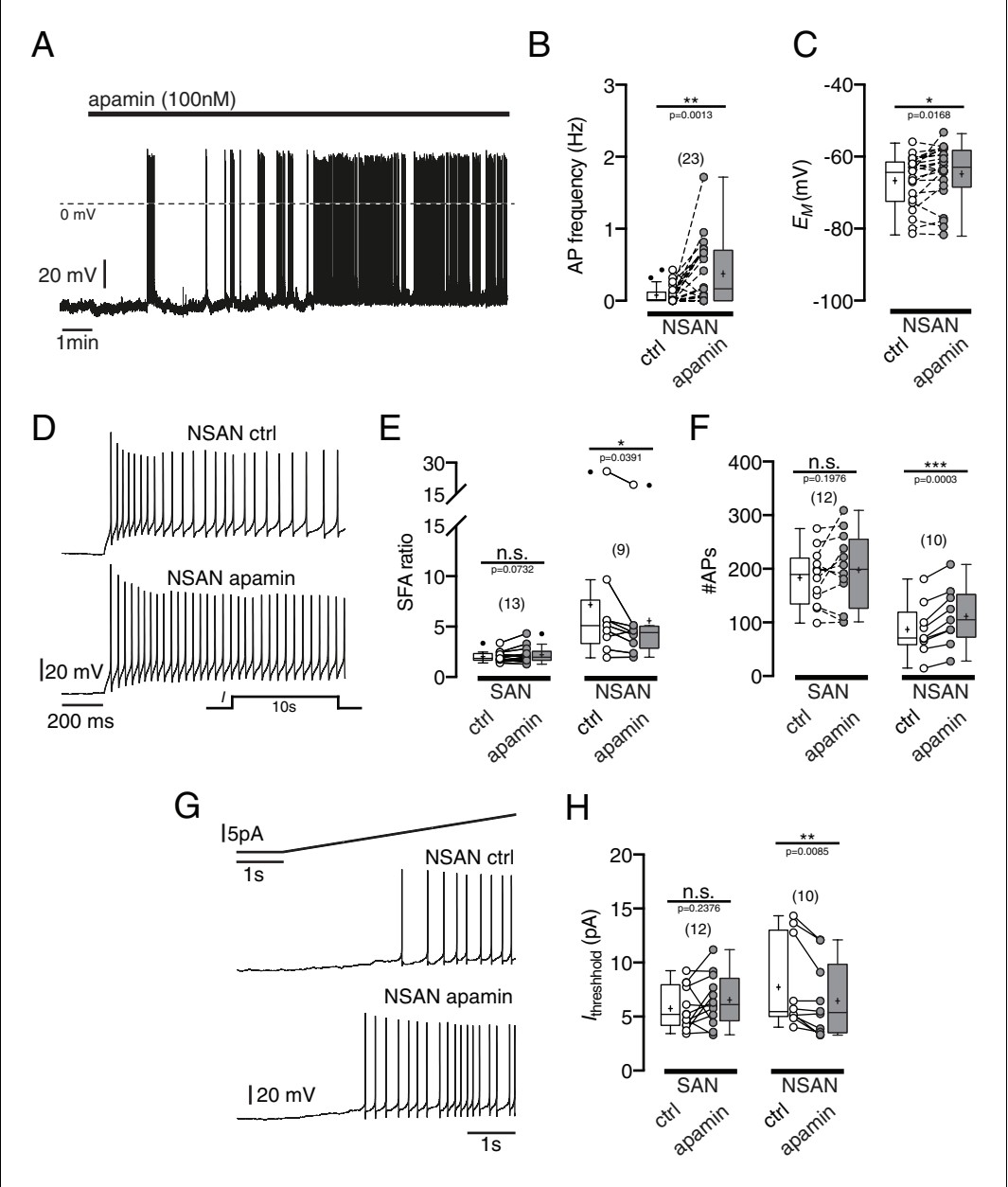

**Figure 9.** Blocking SK currents by apamin changes intrinsic electrophysiological properties of silent POMC neurons. (**A, B, C**) The SK channel blocker apamin (100 nM) induced depolarization and action potential firing. (**A**) A silent POMC neuron depolarized and started to generate action potentials under apamin. Action potential frequency (**B**) and membrane potential (**C**) before and during apamin application. (**D, E, F**) Apamin decreased spike frequency adaptation in silent neurons. (**D**) Original recording of a silent POMC neuron. Apamin reduced the SFA ratio (**E**) and increased the number of action potentials (**F**) during depolarization in silent but not in spontaneously active neurons. (**G, H**) Apamin reduced $I_{threshold}$ only in silent POMC neurons. (**G**) Recording of a silent POMC neuron. (**H**) Quantification of the apamin effect on $I_{threshold}$. SAN, spontaneously active neuron; NSAN, not spontaneously active neuron (F < 0.5 Hz). Paired t-test. *p<0.05, **p<0.01, ***p<0.001.

direct evidence for a diet-dependent deterioration of $Ca^{2+}$ homeostasis in POMC neurons during obesity development.

## Discussion

Satiety-signaling anorexigenic POMC neurons in the ARH play a pivotal role in the regulation of energy homeostasis. While the acute physiological role of various hormones and nutrients and their signaling mechanisms in the CNS are increasingly well understood (*Gao and Horvath, 2007*; *Morton et al., 2006*; *Power, 2012*; *Schwartz and Porte, 2005*), the effects of sustained high caloric intake and obesity on these circuits remain elusive. However, defining diet-associated changes in this network and elucidating their cellular and molecular mechanisms is critical to further unravel the mechanisms leading to increased susceptibility to metabolic disorders. The present study has clearly established that the intrinsic physiological state of anorexigenic POMC neurons is markedly altered in DIO. The starting point for this study was recent work that reported POMC neuron-specific alterations in mitochondrial morphology and decreased mitochondria-ER contacts in DIO by downregulation of MNF2 expression (*Schneeberger et al., 2013*). Mitochondrial shape changes influence the activity of mitochondria and can thus have dramatic effects on cellular signaling. Since mitochondria play a crucial role in intracellular $Ca^{2+}$ handling, we tested if $Ca^{2+}$ handling of POMC neurons is altered in DIO. DIO caused a reduction in $Ca^{2+}$-buffer capacity, a reduction in $Ca^{2+}$ extrusion rate, and an elevation of levels of free intracellular $Ca^{2+}$. These findings are in line with the decreased $Ca^{2+}$ content of mitochondria also found in these experiments, which indicates impaired mitochondrial $Ca^{2+}$ handling. The changes in intracellular $Ca^{2+}$ handling were accompanied by a hyperpolarized membrane potential, a reduction in action potential frequency, and an increase in the number of POMC neurons with no spontaneous activity. Since the recordings were performed in the presence of synaptic blockers, these alterations in electrophysiological activity were not caused by changes in synaptic input, which have been described previously (*Klöckener et al., 2011*; *Newton et al., 2013*).

*Cowley et al. (2001)* reported that leptin depolarizes POMC neurons and increases their firing rate; hence, it is conceivable that DIO-induced reduction in intrinsic activity also contributes to leptin resistance. Former studies had primarily focused on defining alterations in leptin-evoked signaling in POMC neurons, revealing ER stress, reduced leptin-evoked Stat-3 phosphorylation, α-MSH secretion in hypothalamic sections of HFD-fed obese mice (*El-Haschimi et al., 2000*), and increased expression of SOCS-3 or fatty acid-induced activation of inhibitory stress kinases (*Belgardt et al., 2010*; *Ernst et al., 2009*; *Kievit et al., 2006*; *Kleinridders et al., 2009*; *Hosoi et al., 2008*; *Ramírez and Claret, 2015*).

Given the results of these previous studies, it is likely that, from a mechanistic point of view, the decreased activity of POMC neurons reflects the net effect of DIO on multiple pathways. Nevertheless, our experiments, in which altering the intracellular $Ca^{2+}$ concentration could reproduce the electrophysiological properties observed in DIO, identified the intracellular $Ca^{2+}$ concentration as a probable direct link between DIO-induced changes in mitochondrial morphology and function, and the reduction in excitability of POMC neurons. Nevertheless, the downstream targets of $Ca^{2+}$ signals have yet to be revealed. Obvious candidates for mediating the observed changes are $Ca^{2+}$ activated $K^{+}$ channels (*Kaczorowski et al., 1996*; *Kimm et al., 2015*; *McManus, 1991*; *Pedarzani and Stocker, 2008*; *Sah and Davies, 2000*; *Weatherall et al., 2010*, *Li and Bennett, 2007*; *Liu and Herbison, 2008*). This notion is supported by our experiments which show that the SK channel blocker apamin can, at least in part, revers the $Ca^{2+}$ induced decrease in neuronal activity. Nevertheless, there may well be other $Ca^{2+}$-dependent pathways that can reduce neuronal excitability (*Ashcroft and Ashcroft, 1990*; *Ha et al., 2016*).

On the systemic level, the clearly decreased activity of POMC neurons represents a critical reduction in, or even loss of, an important satiety signal, which raises questions regarding the etiology of these dramatic changes. One can assume that POMC neurons initially respond to high-fat feeding with an increase in activity in order to prevent a positive energy balance. Subsequently, chronic overactivation could lead to compensatory adaptations to reestablish normal neuronal baseline activity, as it has been demonstrated in numerous neuronal systems (*Marder and Prinz, 2002*). However, the drastic decrease in neuronal activity of anorexic POMC neurons in a situation of sustained high caloric intake clearly argues against a 'controlled' compensatory mechanism aiming at the reestablishment of normal neuronal responses. Considerable $Ca^{2+}$ entry into POMC neurons during sustained activation by HFD might contribute to the impairment of $Ca^{2+}$ homeostasis (*Suyama et al., 2017*). Thus, it seems more likely that the sustained metabolic stress of HFD-feeding directly impairs $Ca^{2+}$

homeostasis (*Fu et al., 2011*). This notion is in line with previously described alterations in mitochondrial morphology and decreases in mitochondria-ER contacts, caused by downregulation of MNF2 (*Bach et al., 2003*; *Guillet et al., 2011*; *Pich et al., 2005*; *Schneeberger et al., 2013*). This is also interesting in the context of previous work which indicates that an opposite caloric regimen, namely caloric restriction, is an effective way to drastically delay the onset of age related neurodegenerative diseases and age-related impairment of intrinsic neuronal properties (*Hemond and Jaffe, 2005*; *Murchison and Griffith, 2007*). Since several lines of evidence suggest that age-dependent changes in $Ca^{2+}$ homeostasis may increase the susceptibility for age-dependent impairment of neuronal function (*Berridge, 2012*; *Marambaud et al., 2009*; *Toescu and Verkhratsky, 2007*), a compromised $Ca^{2+}$ homeostasis induced by high caloric intake might accelerate aging. Since age-related changes in neuronal activity have also been observed in POMC neurons (*Yang et al., 2012*), the decreased activity observed in our study may be associated with DIO-induced accelerated aging.

From a mechanistic point of view, additional mechanisms have to be taken into account. Since the byproducts of nutrient oxidation are free radicals, and since satiety is associated with high levels of ROS production in POMC neurons, oxidative stress is a likely candidate for modulating mitochondrial function (*Barsukova et al., 2011*). While ROS can act as a physiological signaling molecule at low concentrations (*Brookes et al., 2004*; *Zima and Blatter, 2006*), and while acute increases in ROS levels have been shown to activate POMC neurons (*Diano et al., 2011*), it is not unreasonable to anticipate that sustained high ROS levels as a consequence of high-fat feeding impair mitochondrial function (*Horvath et al., 2009*). At high concentrations, ROS can induce mitochondrial permeability transition, causing mitochondrial $Ca^{2+}$ release and elevation of cytosolic $Ca^{2+}$ levels (*Barsukova et al., 2011*).

In summary, our study shows that DIO dramatically reduces the activity of satiety-mediating POMC neurons. Our experiments suggest that impaired mitochondrial $Ca^{2+}$ handling causes an increase in free cytosolic $Ca^{2+}$, which mediates membrane hyperpolarization. Our study thus provides direct evidence that besides altering hormonal responses, chronic HFD-feeding induces a direct, profound impairment of cell-intrinsic properties, which are of critical importance for neuronal excitability in energy balance-regulating circuits. Further defining the exact molecular pathways leading to impaired $Ca^{2+}$ handling during the development of diet-induced obesity may thus set the ground for novel therapeutic approaches to tackle the current obesity epidemic.

## Materials and methods

### Animals and brain slice preparation

All animal procedures were conducted in compliance with guidelines approved by local government authorities (LANUV NRW, Recklinghausen, Germany) and were in accordance with NIH guidelines. Experiments were performed on brain slices from 18-week-old male POMC-EGFP mice that expressed green fluorescent protein (eGFP) selectively in pro-opiomelanocortin (POMC) neurons (*Cowley et al., 2001*). Animals were kept under standard laboratory conditions, with tap water and chow available ad libitum, on a 12 hr light/dark cycle. Animals were either fed normal chow diet (NCD; Teklad Global Rodent 2018; Harlan, Madison, WI, USA) containing 53.5% carbohydrates, 18.5% protein, and 5.5% fat (12% of calories from fat) or a high-fat diet (HFD; C1057; Altromin, Lage, Germany) containing 32.7% carbohydrates, 20% protein, and 35.5% fat (55.2% of calories from fat) for 12 weeks (starting at an age of 6 weeks). The animals were lightly anesthetized with isoflurane (B506; AbbVie Deutschland GmbH and Co KG, Ludwigshafen, Germany) and subsequently decapitated. Coronal slices (270–300 μm) containing the arcuate nucleus of the hypothalamus (ARH) were cut with a vibration microtome (HM-650 V; Thermo Scientific, Walldorf, Germany) under cold (4°C), carbogenated (95% $O_2$ and 5% $CO_2$), glycerol-based modified artificial cerebrospinal fluid (GaCSF) (*Ye et al., 2006*). The GaCSF contained (in mM): 250 Glycerol, 2.5 KCl, 2 $MgCl_2$, 2 $CaCl_2$, 1.2 $NaH_2PO_4$, 10 HEPES, 21 $NaHCO_3$, and 5 Glucose adjusted to pH 7.2 with NaOH. If not mentioned otherwise, the brain slices were continuously superfused with carbogenated aCSF at a flow rate of ~2 ml·$min^{-1}$. The aCSF contained (in mM): 125 NaCl, 2.5 KCl, 2 $MgCl_2$, 2 $CaCl_2$, 1.2 $NaH_2PO_4$, 21 $NaHCO_3$, 10 HEPES, and 5 Glucose adjusted to pH 7.2 with NaOH. To block synaptic currents, it contained $10^{-4}$ M PTX, $5 \times 10^{-5}$ M D-AP$_5$, and $10^{-5}$ M CNQX.

## Electrophysiology

POMC neurons were recorded at room temperature under current- and voltage-clamp in the perforated patch and whole-cell patch-clamp configuration using an EPC10 patch-clamp amplifier (HEKA, Lambrecht, Germany). Current clamp recordings in the perforated patch configuration were performed using protocols modified from *Horn and Marty (1988)*, *Rae et al. (1991)*, and *Akaike and Harata (1994)*. The recordings were performed with ATP and GTP free pipette solution containing (in mM): 128 K-gluconate, 10 KCl, 10 HEPES, 0.1 EGTA, 2 MgCl$_2$, and adjusted to pH 7.3 with KOH. The patch pipette was tip-filled with internal solution and back-filled with tetraethylrhodamine-dextran (D3308; Invitrogen, Eugene, OR, USA) and amphotericin B (~200 µg·ml$^{-1}$; A4888; Sigma-Aldrich, Taufkirchen, Germany) or nystatin-containing (~200 µg·ml$^{-1}$; N6261; Sigma-Aldrich) internal solution to achieve perforated patch recordings. Whole-cell recordings were performed following the methods of Hamill *et al.* (*Hamill et al., 1981*).

## Spike frequency adaptation

For SFA ratios 10 s depolarizing stimuli were applied from a holding potential of ~−70 mV with initial instantaneous AP frequencies between 30 and 40 Hz. Instantaneous frequencies were plotted (Y in the next equation) over the 10 s time course and fit to a mono-exponential decay equation with $Y_0$ set to the initial instantaneous frequency:

$$Y = (Y_0 - \text{Plateau}) \cdot \exp(-K \cdot T) + \text{Plateau}$$

where Plateau is the asymptotic frequency, K the inverse timeconstant and T the time.

## Ca$^{2+}$ imaging experiments

For Ca$^{2+}$ imaging experiments, slices were superfused with carbogenated aCSF. To block synaptic currents, it contained 10$^{-4}$ M PTX, 5 × 10$^{-5}$ M D-AP5, and 10$^{-5}$ M CNQX. The pipette solution contained (in mM): 135 K-gluconate, 10 KCl, 10 HEPES, 2 MgCl$_2$, and 0.1 fura-2 (pentapotassium salt, F1200, Molecular Probes, OR, USA) adjusted with KOH to pH 7.3.

## Fluorimetric Ca$^{2+}$ measurements

Intracellular Ca$^{2+}$ concentrations and dynamics were measured with the Ca$^{2+}$ indicator fura-2, a ratiometric dye suitable to determine absolute Ca$^{2+}$ concentration after calibration (*Grynkiewicz et al., 1985*). The Ca$^{2+}$-handling parameters were determined as described previously using the 'added buffer' approach (*Neher and Augustine, 1992*) in combination with whole-cell patch-clamp recordings and fast optical imaging. The imaging setup consisted of an Imago/SensiCam CCD camera with a 640 × 480 chip (Till Photonics, Gräfelfing, Germany) and a Polychromator IV (Till Photonics) that was coupled via an optical fiber into the upright microscope. The fura-2-loaded neurons were illuminated during data collection with 340 nm, 360 nm, or 380 nm. Emitted fluorescence was detected through a 440 nm long-pass filter (LP440). Data were acquired as 80 × 60 frames using 8 × 8 on-chip binning. Images were recorded in analog-to-digital units (ADUs) and stored and analyzed as 12-bit grayscale images. For all calculations of kinetics, the mean values of ADUs within regions of interest (ROIs) from the center of the cell bodies were used.

Free intracellular Ca$^{2+}$ concentrations were determined as in (*Grynkiewicz et al., 1985*):

$$\left[ Ca^{2+} \right]_i = K_{d,Fura} \frac{F_{380,min}}{F_{380,max}} \cdot \frac{(R - R_{min})}{(R_{max} - R)} \tag{1}$$

$\left[ Ca^{2+} \right]_i$ is the free intracellular Ca$^{2+}$ concentration for the background-subtracted fluorescence ratio $R$ from 340 nm and 380 nm excitation. $R_{min}$ and $R_{max}$ are the ratios at a Ca$^{2+}$ concentration of virtually 0 M and at saturating Ca$^{2+}$ concentrations, respectively. $K_{d,Fura}$ is the dissociation constant of fura-2. $F_{380,min}/F_{380,max}$ is the ratio between the emitted fluorescence of Ca$^{2+}$-free dye and the emitted fluorescence of Ca$^{2+}$-saturated dye at 380 nm excitation, reflecting the dynamic range of the indicator. The term $K_{d,Fura} \cdot (F_{380,min}/F_{380,max})$ is substituted with the effective dissociation constant $K_{d,Fura,eff}$, which is independent of the dye concentration and specific for each experimental setup:

$$K_{\mathrm{d,Fura,eff}} = \left[\mathrm{Ca}^{2+}\right]_{\mathrm{i}} \frac{(R_{\max} - R)}{(R - R_{\min})} \tag{2}$$

The free $\mathrm{Ca}^{2+}$ concentrations in the calibration solutions were determined using a $\mathrm{Ca}^{2+}$-selective electrode (*Kay et al., 2008*; *McGuigan et al., 1991*). Calibration solutions contained (in mM): $R_{\max}$: 140 KCl, 2.5 KOH, 15 NaCl, 1 MgCl$_2$, 5 HEPES, 10 CaCl$_2$, and 0.05 fura-2; $R_{\min}$: 129.5 KCl, 13 KOH, 15 NaCl, 1 MgCl$_2$, 5 HEPES, 4 EGTA, and 0.05 fura-2; $R_{\mathrm{def}}$: 129.5 KCl, 13 KOH, 10.3 NaCl, 4.7 NaOH, 1 MgCl$_2$, 5 HEPES, 4 EGTA, 2.7 CaCl$_2$, and 0.05 fura-2, yielding a free $\mathrm{Ca}^{2+}$ concentration of 0.35 μM. All solutions were adjusted to pH 7.2 with HCl.

$\mathrm{Ca}^{2+}$-handling parameters were determined by using the 'added buffer' approach originally introduced by Neher and Augustine (*Neher and Augustine, 1992*). This method is based on a linear, single compartment model, with the rationale that for measurements of intracellular $\mathrm{Ca}^{2+}$ concentrations with $\mathrm{Ca}^{2+}$ chelator-based indicators, the amplitude and time course of the signals are dependent on the concentration of the $\mathrm{Ca}^{2+}$ indicator (here: fura$-$2), which acts as an exogenous $\mathrm{Ca}^{2+}$ buffer (B), and the endogenous buffer (S). The ability of the experimentally introduced exogenous buffer to bind $\mathrm{Ca}^{2+}$ is described by its $\mathrm{Ca}^{2+}$-binding ratio that is defined as the ratio of the change in buffer-bound $\mathrm{Ca}^{2+}$ over the change in free $\mathrm{Ca}^{2+}$:

$$\kappa_{\mathrm{B}} = \frac{d[\mathrm{BCa}]}{d\left[\mathrm{Ca}^{2+}\right]_{\mathrm{i}}} = \frac{[\mathrm{B_T}]K_{\mathrm{d,B}}}{\left(\left[\mathrm{Ca}^{2+}\right]_{\mathrm{i}} + K_{\mathrm{d,B}}\right)^2} \tag{3}$$

$[\mathrm{B_T}]$ is the total concentration of the exogenous buffer $B$, and $K_{\mathrm{d,B}}$ is its dissociation constant for $\mathrm{Ca}^{2+}$. In this model, the decay time constant ($\tau_{\mathrm{transient}}$) of a $\mathrm{Ca}^{2+}$ transient induced by a brief $\mathrm{Ca}^{2+}$ influx is described as:

$$\tau_{\mathrm{transient}} = \frac{1 + \kappa_{\mathrm{B}} + \kappa_{\mathrm{S}}}{\gamma - \frac{(1 + \kappa_{\mathrm{S}})}{\tau_{\mathrm{loading}}}} \tag{4}$$

$\tau_{\mathrm{transient}}$ is proportional to $\kappa_B$, and a linear fit to the data has its negative x-axis intercept at $1 + \kappa_S$, yielding the endogenous $\mathrm{Ca}^{2+}$-binding ratio of the cell. The slope of this fit is the inverse of the $\mathrm{Ca}^{2+}$ extrusion rate $\gamma$. The y-axis intercept yields the time constant $\tau_{\mathrm{endo}}$ for the decay of the $\mathrm{Ca}^{2+}$ transient as it would appear in the cell without exogenous buffer. $\tau_{\mathrm{transient}}$ values were plotted as a function of $\kappa_{\mathrm{B}}$ values and fit with a linear function $Y = \beta_0 + \beta_1 x$, using the 'R function' lm (R Development Core Team [2009]). To estimate the variance of the $\mathrm{Ca}^{2+}$-handling parameters, we used the bootstrap method (*Pippow et al., 2009*) implemented in the boot library in R (fixed-x resampling, 1000 bootstrap samples, boot: Bootstrap R Functions, R package version 1.2–36), which provided bootstrap distributions (n = 1000) for each of the parameters. The distributions were log-transformed to bring them closer to a Gaussian. To determine differences in means between the different cell types, unpaired t-tests were performed.

To monitor calcium release from mitochondria into the cytoplasm, the calcium indicator fura-2 was AM-loaded into POMC neurons by incubating brain slices in carbogenated aCSF containing 10 μM fura-2 (fura-2 AM, F1221, Molecular Probes; 24°C for 50–60 min). During the recordings, slices were superfused with aCSF additionally containing 250 μM sulfinpyrazone (S9509, Sigma-Aldrich), to inhibit fura-2 sequestration (*Di Virgilio et al., 1990*). $\mathrm{Ca}^{2+}$ release from mitochondria was induced by bath application of 2 μM carbonyl cyanide p-(trifluoromethoxy) phenylhydrazone (FCCP, Sigma-Aldrich) at a flow rate of 3 ml·min$^{-1}$. Since the background fluorescence is not clearly determinable in AM-loaded cells, no background was subtracted, and instead of reporting absolute changes in free intracellular $\mathrm{Ca}^{2+}$, changes in calcium concentration were given as changes in the fura-2 ratio (340/380 nm) (*Bergmann and Keller, 2004*; *Jaiswal et al., 2009*). For better visualization and comparison of the time courses of the fura-2 ratio between the two populations, we normalized the fluorescence ratio to the baseline. Analysis was performed off-line using IGOR Pro 6 (Wavemetrics, Lake Oswego, OR, USA).

## Tools for data analysis and statistics

Electrophysiological data were analyzed using Pulse (version 8.63, HEKA-Elektronik) and Igor Pro six software (Wavemetrics, including the Patcher's Power Tools plug-in: http://www.mpibpc.mpg.de/

groups/neher/index.php?page=software). All calculations for the determination of EGTA purity, its dissociation constant, and the free $Ca^{2+}$ concentrations in the calibration solutions were performed in R (R Development Core Team [2009], http://www.R-project.org). All functions that were used to fit the $Ca^{2+}$ buffering-related data with a linearized one-compartment model were implemented in R (R Development Core Team [2009]. If not stated otherwise, all calculated values are expressed as means ± SEM (standard error of the mean). For pairwise comparisons of dependent and independent normal distributions, paired and unpaired t-tests were used, respectively. For pairwise comparisons of independent, not normal distributions, Mann-Whitney U-tests were used. For multiple comparisons, ANOVA with Bonferroni post hoc analysis was performed. Tests were executed using GraphPad Prism 5 (GraphPad Software Inc., La Jolla, CA, USA). A significance level of 0.05 was accepted for all tests. Significance levels were: $*p<0.05$, $**p<0.01$, $***p<0.001$. In all figures, n-values are given in brackets. Exact p-values are reported if $p>0.0001$.

## Acknowledgements
We would like to thank Helmut Wratil for excellent technical assistance.

## Additional information

### Funding

| Funder | Grant reference number | Author |
| --- | --- | --- |
| Boehringer Ingelheim Fonds | Fellowship | Moritz Paehler |
| CECAD | | Jens C Brüning Peter Kloppenburg |
| Max-Planck-Gesellschaft | | Jens C Brüning |
| Deutsche Forschungsgemeinschaft | TR-SFB 134/TP A03 | Peter Kloppenburg |
| Deutsche Forschungsgemeinschaft | SFB 1218/TP B07 | Peter Kloppenburg |

The funders had no role in study design, data collection and interpretation, or the decision to submit the work for publication.

### Author contributions
LP, Conceptualization, Data curation, Formal analysis, Investigation, Writing—original draft, Writing—review and editing; AP, Conceptualization, Data curation, Formal analysis, Investigation, Writing—review and editing; SH, Formal analysis, Investigation, Writing—review and editing; MP, Formal analysis, Investigation; ACK, AH, Investigation; CP, Software, Formal analysis, Investigation; JCB, Resources, Validation; PK, Conceptualization, Resources, Formal analysis, Supervision, Funding acquisition, Writing—original draft, Project administration, Writing—review and editing

### Author ORCIDs
Lars Paeger, http://orcid.org/0000-0001-8716-3483
Christophe Pouzat, http://orcid.org/0000-0002-2844-8099
Peter Kloppenburg, http://orcid.org/0000-0002-4554-404X

### Ethics
Animal experimentation: All animal procedures were conducted in compliance with guidelines approved by local government authorities (8.87-51.05.20.09.208, 84-02.05.20.12.254, 8.87-50.10.31.08.279, 84-02.04.2014.A511) (LANUV NRW, Recklinghausen, Germany) and were in accordance with NIH guidelines.

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
