## [Decision Letter]

Thank you for submitting your article "Energy imbalance alters Ca-handling and excitability of POMC neurons" for consideration by *eLife*. Your article has been reviewed by three peer reviewers, and the evaluation has been overseen by a Reviewing Editor and Richard Aldrich as the Senior Editor. The reviewers have opted to remain anonymous. The reviewers have discussed the reviews with one another and the Reviewing Editor has drafted this decision to help you prepare a revised submission.

Summary:

Paeger and colleague provide an interesting series of ex vivo experiments investigating the effect of diet-induced obesity (DIO) on POMC neuron excitability and calcium handling. This work builds on previous reports demonstrating decreased mitochondria endoplasmic reticulum contacts in POMC neurons in DIO models, reduced POMC activity with age-associated obesity and high fat diet rewiring of inputs to POMC neurons. The authors find that high-fat-diet increases intracellular calcium and reduces electrical excitability of anorexigenic POMC neurons in the hypothalamic arcuate nucleus. Through a series of classic and elegant experiments examining calcium buffering and extrusion, they show that the disruption in calcium in associated with DIO-induced reduction in cytosolic calcium buffering and extrusion, as well as by reduced capacity for mitochondrial calcium uptake. Finally they show that increasing intracellular calcium (by increasing extracellular calcium) is sufficient to recapitulate DIO-associated changes in POMC electrical properties.

Essential revisions:

A key point is the mechanisms linking calcium and spiking/Vm in POMC neurons. Obvious candidate are calcium-activated K channels. The authors should test whether the relation between extracellular calcium and SFA/Vm is mediated by these channels, e.g. by using K/Ca blockers (apamin, iberiotoxin or similar), in conjunction with I-V analysis of the membrane.

A previous report suggested that "although average POMC neuronal volume does not change with the length of chow diet or HFD exposure (2 and 6 month), both HFD exposure and length of diet exposure alters the normal distribution of POMC cell volumes." Might the, "increased level of free intracellular Ca^2+^ in POMC neurons of DIO compared to control mice" in the current study be less surprising if the volume of POMC neurons examined also increased in response to HFD? Have the authors accounted for possible changes in cell number or volume?

Are the results related to obesity, diet or something else entirely? The impact of this work would be substantially increased if the authors could identify a trigger for the change in POMC activity. For example, the Materials and methods suggest that 15-20 week old mice (fed chow or HFD) were used in this study. However, mice were fed 15 weeks of HFD before experimentation. How long were mice on chow diet or HFD diet? Assuming a weaning age of 3 weeks, mice fed 15 weeks HFD would be at least 18-20 weeks of age; while chow fed mice could be 15-20 weeks. This is an important point given that a previous report suggested that POMC neurons are susceptible to age dependent changes in synaptic and cell intrinsic properties. A critical time for these changes to occur is between 15-30 weeks of age. It's unclear why the authors have chosen a time point in this range to test DIO-induced changes in cellular properties. The authors should consider using younger mice or narrowing the range of ages used in the current study to eliminate this concern.

Possibly considering shorter exposure periods to HFD might help to delineate effects of the HFD versus the various metabolic pathologies contributing to obesity and diabetes. For example, are these changes driven by alterations in insulin, leptin, diet, adiposity, etc.? Jan's report in Neuron 5 years ago (75, 425-436, 2012) illustrates that age-associated obesity hyperpolarizes POMC neurons. This earlier paper reduces the novelty of the current report assessing POMC activity in another form of obesity, DIO. Jan compared mice at 1 month, 6 months, 12 months and 18 months and found that by 6 months, POMC neurons were hyperpolarized compared to 1 month of age. In the current report, the authors evaluated mice at 15-20 weeks. One interpretation of the current data could be that degree of obesity accelerates changes in POMC neuron activity. Another report indicates that only 3 days exposure of high fat diet re-wires inputs onto POMC neurons (Benani et al., The Journal of Neuroscience, 2012 • 32(35):11970 -11979), and this is associated with an increase in fat mass within this short time period. If the present authors included mice at different time points of high fat diet exposure, with or without pair-feeding to chow controls, this could tease a possible explanation for changes in POMC activity related to diet or adiposity.

[Editors' note: further revisions were requested prior to acceptance, as described below.]

Thank you for resubmitting your work entitled "Energy imbalance alters Ca^2+^ handling and excitability of POMC neurons" for further consideration at *eLife*. Your revised article has been favorably evaluated by Richard Aldrich (Senior Editor), a Reviewing Editor, and two reviewers.

The manuscript has been improved but there are some remaining issues that need to be addressed before acceptance.

As outlined below, some textual changes are needed. In particular, the comment regarding discussion of the DIO and the Ca activated K channel.

This is a revised study on the alterations in intrinsic cellular properties of Pomc on a chow or HFD diet. The authors demonstrate a role for cellular and mitochondrial calcium dynamics in activity of Pomc neurons. The authors have added data, including the involvement of Ca^2+^ activated K channels in their responses. However, it's unclear if these data are relevant to the observed DIO effects described.

1) The added data examining Ca^2+^ activated K^+^ channels in the excitability of Pomc neurons seems misleading. The title of the section indicates, "SK channels contribute to decreased excitability of POMC neurons in DIO". However the authors explain that since DIO increases the Ca^2+^ resting level and hyperpolarized the membrane potential, they used an artificial model of recording with high [Ca^2+^]_i_ rather than recording Pomc neurons from DIO mice. While I understand the utility of being able to control the calcium levels, this leaves the question if SK channels contribute to the inactivity of Pomc neurons in DIO? Currently, the authors have only shown that high Ca^2+^ contributes to SK activity (which is not surprising).

2) It's surprising that the authors have not referenced a previous report demonstrating a HFD-induced suppression of POMC neuron activity as such (Diano et al., 2011).

*Reviewer #2:*

The authors have carefully and thoroughly performed additional experimental analyses that I suggested (ion channel blockade to investigate mechanisms), with further important results. Altogether this is an interesting and original paper, and I am happy to recommend it for publication.

---

## [Author Response]

*Essential revisions:*

*A key point is the mechanisms linking calcium and spiking/Vm in POMC neurons. Obvious candidate are calcium-activated K channels. The authors should test whether the relation between extracellular calcium and SFA/Vm is mediated by these channels, e.g. by using K/Ca blockers (apamin, iberiotoxin or similar), in conjunction with I-V analysis of the membrane.*

We agree with the reviewers that it is important to investigate and understand the mechanisms that link DIO induced changes in the intracellular Ca^2+^ concentration and the DIO induced changes in intrinsic electrophysiological properties. Following the suggestions of the reviewers, we have tested whether Ca^2+^ activated K^+^ channels, which are obvious candidates, are involved. To directly address this link, we have performed numerous additional electrophysiological experiments. The results of these critical new experiments are shown in the new Figure 9, and stated and discussed in the fifth, sixth and seventh paragraphs of the subsection “Spike frequency adaptation is increased in silent POMC neurons”.

In this context is important to consider that there may well be other Ca^2+^ dependent pathways that can reduce the neural excitability, as we point out in the Discussion (third paragraph).

*A previous report suggested that "although average POMC neuronal volume does not change with the length of chow diet or HFD exposure (2 and 6 month), both HFD exposure and length of diet exposure alters the normal distribution of POMC cell volumes." Might the, "increased level of free intracellular Ca^2+^ in POMC neurons of DIO compared to control mice" in the current study be less surprising if the volume of POMC neurons examined also increased in response to HFD? Have the authors accounted for possible changes in cell number or volume?*

We thank the reviewer for pointing out this interesting paper that investigated the number and volume of POMC neurons. While Lemus et al. (Endocrinology 156: 1701-1713, 2015) found that HFD exposure of 2 and 6 months did not change the mean number and mean volume of POMC neurons, a volume distribution analysis revealed a significant reduction in POMC cell number at small cell volumes. However, in our opinion it would be extremely speculative to construct a mechanistic link between our current work and the paper from Lemus et al. at this point in time. Therefore, we did not include this topic in our Discussion.

*Are the results related to obesity, diet or something else entirely? The impact of this work would be substantially increased if the authors could identify a trigger for the change in POMC activity. For example, the Materials and methods suggest that 15-20 week old mice (fed chow or HFD) were used in this study. However, mice were fed 15 weeks of HFD before experimentation. How long were mice on chow diet or HFD diet? Assuming a weaning age of 3 weeks, mice fed 15 weeks HFD would be at least 18-20 weeks of age; while chow fed mice could be 15-20 weeks. This is an important point given that a previous report suggested that POMC neurons are susceptible to age dependent changes in synaptic and cell intrinsic properties. A critical time for these changes to occur is between 15-30 weeks of age. It's unclear why the authors have chosen a time point in this range to test DIO-induced changes in cellular properties. The authors should consider using younger mice or narrowing the range of ages used in the current study to eliminate this concern.*

First of all, we want to apologize for not having stated the experimental conditions regarding age- and feeding status of the experimental animals in great enough detail. We absolutely agree with the reviewers that this is critical information. This information is now clearly stated in ‘Results’ (subsection “High- fat diet decreases activity of anorexigenic POMC -neurons”, first paragraph), and in ‘Materials and Methods’ (subsection “Animals and brain slice preparation”.

All electro- and optophysiological data that are included in the revised version were collected from 18 weeks old animals. The mice were fed NCD or HFD for 12 weeks, starting at an age of 6 weeks. This regimen was chosen to exactly match the experiments of Schneeberger et al. (Cell 155, 172-187, 2013), in which they observed loss of mitochondria-ER contacts and changes in mitochondrial network complexity. Their work served as the direct basis for our study and in particular the experimental conditions.

In the previous version of the manuscript we had included electrophysiological data (only in Figure 1) from animals of similar but not identical age and feeding status. These data are now omitted from Figure 1.

*Possibly considering shorter exposure periods to HFD might help to delineate effects of the HFD versus the various metabolic pathologies contributing to obesity and diabetes. For example, are these changes driven by alterations in insulin, leptin, diet, adiposity, etc.? Jan's report in Neuron 5 years ago (75, 425-436, 2012) illustrates that age-associated obesity hyperpolarizes POMC neurons. This earlier paper reduces the novelty of the current report assessing POMC activity in another form of obesity, DIO. Jan compared mice at 1 month, 6 months, 12 months and 18 months and found that by 6 months, POMC neurons were hyperpolarized compared to 1 month of age. In the current report, the authors evaluated mice at 15-20 weeks. One interpretation of the current data could be that degree of obesity accelerates changes in POMC neuron activity. Another report indicates that only 3 days exposure of high fat diet re-wires inputs onto POMC neurons (Benani et al., The Journal of Neuroscience, 2012 • 32(35):11970 -11979), and this is associated with an increase in fat mass within this short time period. If the present authors included mice at different time points of high fat diet exposure, with or without pair-feeding to chow controls, this could tease a possible explanation for changes in POMC activity related to diet or adiposity.*

We agree with the reviewers, that the suggested experiments (using different periods of exposure to HFD) will provide important insights into the cellular events that trigger the change in mitochondrial shape and function. Nevertheless, particularly in light of the extensive emphasis of the current manuscript on the link between (mitochondrial) Ca^2+^ handling and the electrophysiological properties, we hope that the reviewers agree that these experiments will be subject to a follow-up study and are beyond the scope of this already extensive study.

In our opinion, the paper from Yang et al. (Neuron 75, 425-436, 2012) does not reduce the novelty from our current report. Our understanding is that Yang et al. did not investigate the effect of HFD on the activity of POMC neuron. They found an age dependent decrease in POMC activity, and suggest that this causes an age-related increase in body weight. Nevertheless, this paper is now cited in a different context (age dependent effects) in the fourth paragraph of the Discussion.

[Editors' note: further revisions were requested prior to acceptance, as described below.]

*As outlined below, some textual changes are needed. In particular, the comment regarding discussion of the DIO and the Ca activated K channel.*

*This is a revised study on the alterations in intrinsic cellular properties of Pomc on a chow or HFD diet. The authors demonstrate a role for cellular and mitochondrial calcium dynamics in activity of Pomc neurons. The authors have added data, including the involvement of Ca^2+^ activated K channels in their responses. However, it's unclear if these data are relevant to the observed DIO effects described.*

*1) The added data examining Ca^2+^ activated K^+^ channels in the excitability of Pomc neurons seems misleading. The title of the section indicates, "SK channels contribute to decreased excitability of POMC neurons in DIO". However the authors explain that since DIO increases the Ca^2+^ resting level and hyperpolarized the membrane potential, they used an artificial model of recording with high [Ca^2+^]_i_ rather than recording Pomc neurons from DIO mice. While I understand the utility of being able to control the calcium levels, this leaves the question if SK channels contribute to the inactivity of Pomc neurons in DIO? Currently, the authors have only shown that high Ca^2+^ contributes to SK activity (which is not surprising).*

We apologize for apparently not having been clear enough in describing the experimental conditions in the experiments that examine the role of Ca^2+^ activated K^+^ channels. In these experiments we used normal extracellular Ca^2+^ concentrations and did not directly manipulate the [Ca^2+^]_i_. Since it was never stated that we directly manipulate [Ca^2+^]_i_ in these experiments, we assume that the confusion was caused by the sentence in the third paragraph of the subsection “Spike frequency adaptation is increased in silent POMC neurons”. This sentence was modified and we hope that we have clarified this issue.

*2) It's surprising that the authors have not referenced a previous report demonstrating a HFD-induced suppression of POMC neuron activity as such (Diano et al., 2011).*

Diano et al., 2011 has been referenced in the Discussion (fifth paragraph). Since we agree with the reviewer that this is a great paper, we now reference it a second time in the Introduction (third paragraph).